# The Development of Electronic Health and Artificial Intelligence in Surgery after the SARS-CoV-2 Pandemic—A Scoping Review

**DOI:** 10.3390/jcm10204789

**Published:** 2021-10-19

**Authors:** Stephanie Taha-Mehlitz, Ahmad Hendie, Anas Taha

**Affiliations:** 1Clarunis, University Center for Gastrointestinal and Liver Diseases, St. Clara Hospital and University Hospital, 4002 Basel, Switzerland; Stephanie.taha@clarunis.ch; 2Department of Computer Engineering, McGill University, Montreal, QC H3A 0C6, Canada; ahmad.hendie@mail.mcgill.ca; 3Department of Biomedical Engineering, Faculty of Medicine, University of Basel, 4321 Allschwil, Switzerland

**Keywords:** surgery, pandemic, SARS-CoV-2, COVID-19, eHealth, artificial intelligence

## Abstract

Background: SARS-CoV-2 has significantly transformed the healthcare environment, and it has triggered the development of electronic health and artificial intelligence mechanisms, for instance. In this overview, we concentrated on enhancing the two concepts in surgery after the pandemic, and we examined the factors on a global scale. Objective: The primary goal of this scoping review is to elaborate on how surgeons have used eHealth and AI before; during; and after the current global pandemic. More specifically, this review focuses on the empowerment of the concepts of electronic health and artificial intelligence after the pandemic; which mainly depend on the efforts of countries to advance the notions of surgery. Design: The use of an online search engine was the most applied method. The publication years of all the studies included in the study ranged from 2013 to 2021. Out of the reviewed studies; forty-four qualified for inclusion in the review. Discussion: We evaluated the prevalence of the concepts in different continents such as the United States; Europe; Asia; the Middle East; and Africa. Our research reveals that the success of eHealth and artificial intelligence adoption primarily depends on the efforts of countries to advance the notions in surgery. Conclusions: The study’s primary limitation is insufficient information on eHealth and artificial intelligence concepts; particularly in developing nations. Future research should focus on establishing methods of handling eHealth and AI challenges around confidentiality and data security.

## 1. Introduction

Boogerd et al. state that electronic health has gained popularity over the last ten years, but scholars are yet to reach a consensus concerning the definition of the term [1]. Baker, Gustafson, and Shah stated that the transformations in the eHealth domain had challenged the way researchers conduct studies, which has led to the establishment of efficient publications [2]. Furthermore, Meier, Fitzgerald, and Smith argue that electronic health allows clinicians to deliver healthcare services via the Internet and other related resources [3]. Van der Kleij et al. argue that eHealth represents an enhanced state of mind and attitude towards advancing healthcare delivery through the use of information and technical messages [4]. Artificial intelligence has received immense demand in the healthcare domain, whereby practitioners and consumers want AI machines that perform well and enhance transparency and trustworthiness [5]. According to Hamet and Tremblay, people in the medical sector can apply AI to numerous items, including human biology, robotics, and medical statistics [6].

Amisha, Pathania, and Rathaur posit that AI falls into two broad categories in medicine, virtual and physical [7]. Despite the tremendous progress made by artificial intelligence in previous years, the scattered improvement does not offer a cohesive structure for an observer [8]. Nevertheless, Buch, Ahmed, and Maruthappu insist that the future of AI application in medicine is bright since the concept will allow the extraction of critical data from patients’ electrical footprints [9].

### 1.1. Rationale

This research contributes to the surgical field by equipping surgeons with relevant information about the eHealth and artificial intelligence area of improvement. The findings of this study will further contribute to future studies by providing future researchers with data to enhance or critique.

### 1.2. Objectives

The primary goal of this scoping review is to elaborate on how surgeons have used eHealth and AI before and after the current pandemic. In other words, this review focuses on the empowerment of the concepts of electronic health and artificial intelligence after the pandemic, which mainly depend on the efforts of countries to advance the notions of surgery. The selection of the reviewed papers occurred by incorporating inclusion and exclusion criteria to guarantee the validity and reliability of gathered information. For example, the reviewed papers’ publication years ranged from 2013 to 2021. Further, the literature review approach integrated into this study was the PRISMA method for scoping review. The research question developed was: What does the literature say about the current situation and future development chances of eHealth and artificial intelligence in surgery globally?

## 2. Methods

The drafting of our protocol occurred via the PRISMA-ScR model that details the critical reporting items to integrate into a study. The literature included in this study focused on the notions of eHealth and AI in surgery. In addition, all journal articles had to be peer-reviewed and published from 2013 to 2021 for inclusion. Our study excluded articles published earlier than the stated dates and focused on eHealth and AI in other sectors other than surgery. The online search engine adopted for this study was GoogleScholar, whereby team discussions facilitated the refining of established results. The search strategy included keywords such as “eHealth in surgery” and “artificial intelligence in surgery” in the search engine. All the reviewers in this study comprehensively evaluated the potential sources by assessing the present data to guarantee relevancy. The reviewers collaborated to identify challenges and solve them. The abstracted data utilized for this study depended on multiple characteristics such as the targeted population of the research and the generated results. The nature of our review permitted us to categorize the collected data based on the countries evaluated.

## 3. Results

The flowchart (Figure 1) below demonstrates how the sources adopted were selected.

The majority of sources adopted for the study were peer-reviewed journal articles acquired from Google Scholar. The researcher further incorporated two website articles.

## 4. Discussion

### 4.1. Development of Electronic Health and Artificial Intelligence in Surgery after the SARS-CoV-2 Pandemic Globally

The onset of the SARS-CoV-2 pandemic has dramatically transformed healthcare systems globally. For instance, the outbreak has prompted the realization that there is a vast connection between eHealth, artificial intelligence, and digital inequalities [10]. These connections imply that the elderly and the impoverished individuals in society are more likely to miss out on the benefits of eHealth globally [10]. Another study by Sockalingam, Leung, and Cassin showcased that the pandemic has increased eHealth use as physicians use the technology to reach out to patients and deal with various distresses after a surgical procedure [11]. A review by Tsopra et al. indicated that developing eHealth and artificial intelligence in surgery after SARS-CoV-2 in the African continent is likely to encounter numerous challenges due to the differences in healthcare guidelines given for different countries [12]. In addition, the authors posit that it is necessary for medical studies in the surgical field to involve general practitioners in the data collection process [12]. Many hospitals globally have had to adopt measures that minimize the risk of hospital staff and patients contracting SARS-CoV-2. For instance, in facial surgeries, doctors have mostly prioritized injuries on critical locations such as the eyelids and deferred cases involving temporomandibular pathologies [13]. The issues on the deferred list have had to rely on eHealth to meet their various medical requirements, hence presenting a chance for the further development of the concept.

Moreover, Melstrom et al. pointed out a growing need for artificial intelligence techniques during the pandemic in predicting surgical outcomes through the available data sets [14]. Other than that, the authors insist that the use of eHealth and AI in surgery permits the derivation of quality interventions that lead to optimum outcomes among surgery oncology patients [14]. Another study by Zemmar, Lozano, and Nelson revealed that the use of robotics during the pandemic has spiraled in the surgical domain due to many medical professionals contracting the virus and the unavailability of sufficient personal protective gear [15]. Nonetheless, there is still room for improvement since hospitals can use AI to perform tasks such as greeting patients when there is an outbreak, providing disinfectants, and distributing personal protective equipment globally [15]. For example, in Canada, AI use was evident whereby BlueDot, a company based in the region, used artificial intelligence to predict the onset of SARS-CoV-2 in Wuhan.

Additionally, Bhaskar et al. proclaimed that the association between eHealth and artificial intelligence is crucial to understand as it will help practitioners establish methods of dealing with future pandemics [16]. Similarly, Khalsa et al. stated that the adoption of electronic health in the cardiac surgical domain had created room for integrating artificial intelligence into the sector [17]. However, the authors pointed out that the incorporation of AI in surgery has numerous drawbacks primarily associated with patient safety [17]. For instance, the association between robotic procedures and patient death has prevailed, despite arguments that physicians’ inexperience with the robots might have contributed to the occurrences. Nevertheless, AI can supplement the surgical field after the pandemic by replacing traditional procedures with more efficient systems. Ahuja and Nair add to this point by insisting on the positive attributes of AI, such as self-reported analysis, the management of drug infusions, and image recognition [18]. Furthermore, AI algorithms have an accuracy rate of 95.2%, similar to that of humans, only that machines solve complicated relations while using enhanced judgment [18]. In Addenbrooke hospital in Cambridge, England, the integration of artificial intelligence was evident, where the hospital used AI tools to predict the oxygen needs of various SARS-CoV-2 patients globally [19]. Therefore, this situation implies that medical practitioners and researchers will develop eHealth and AI systems after the pandemic is over.

### 4.2. Development of Electronic Health and Artificial Intelligence in Surgery after the SARS-CoV-2 Pandemic in the United States

The history of electronic health and artificial intelligence in surgery in the United States dated to 1996 when a French patient underwent a laparoscopic cholecystectomy virtually with a surgeon located in New York [15]. Zemmar, Lozano, and Nelson further stated that the technology adopted in the United States allows patients to undergo surgery without contacting the doctor [13]. For example, the first patient of SARS-CoV-2 underwent isolation with a robot equipped with a camera, microphone, and a stethoscope [15]. A review conducted by Messiah et al. on the application of eHealth in metabolic and bariatric surgery revealed that electronic health strategies effectively allowed post-metabolic bariatric surgery weight loss and enhanced health outcomes [20]. However, the authors noted that most of the studies reviewed had study design limitations; therefore, it was hard to rely on the results [20]. More importantly, Bokolo insisted that the American Medical Association has established many resources to enable physicians to acquire relevant advice via electronic health [21]. In addition, California passed a bill before the pandemic requiring practitioners to adopt virtual technologies and eliminate the various barriers in the reimbursement of Medicaid for communal medical centers.

Amid the pandemic, the United States has further integrated the “Coronavirus Preparedness and Response Supplemental Appropriations Act, 2020” with the primary goal of waiving and modifying the various restriction to eHealth visible in the Medicare package [21]. According to Alonso et al., most of the studies (42%) on electronic health and artificial intelligence during the pandemic focus on the United States, thus implying that the country is at the forefront of applying the concepts in various medical procedures such as surgery [22]. Another article by Feizi et al. revealed that in the United States, the use of AI is evident during the pandemic via the Corpath robotic arm, whose primary goal is to perform coronary interventions, especially in instances where professionals are dealing with SARS-CoV-2 patients [23]. Haleem et al. reported that the prominence of AI application in cardiological surgery has gone up during the pandemic and has led to numerous benefits such as predicting and diagnosing heart illness [24]. Furthermore, artificial intelligence has reduced the time and cost of surgery in the United States while encouraging the successful undertaking of precise surgeries [24]. In neurosurgery, the lack of good AI tools reduced the number of operations [25]. The urgency of neurology surgeries necessitates the development of sound and efficient AI systems after the SARS-CoV-2 era.

A study carried out by Salman et al. in Washington revealed that deep learning went up during the pandemic, whereby surgical departments applied the concept to reduce human contact [26]. The authors also developed a model of artificial intelligence that was one hundred percent accurate in detecting SARS-CoV-2 [26]. Other than that, Secinaro et al. argued that many hospitals in the United States had adopted robotic-assisted surgery in numerous fields such as colorectal, orthopedic, and neural surgeries [27]. Consequently, studies concentrating on the importance of artificial intelligence in surgery have increased over the years, with the highest number witnessed during the SARS-CoV-2 pandemic, as displayed in Figure 2.

### 4.3. Development of Electronic Health and Artificial Intelligence in Surgery after the SARS-CoV-2 Pandemic in Europe

In Europe, the adoption of electronic health and artificial intelligence in surgery has gone up during the pandemic. For example, the Society of European Robotic Gynecological Surgery presented guidelines to doctors that promoted the integration of robot-assisted surgeries to minimize the risks of infection brought about by open surgery [16]. In addition, the European Association of Urology released guidelines warning physicians to manage smoke dispersion in robotic surgery [16]. These facts prove that the development of electronic health and AI in surgery after the SARS-CoV-2 in Europe is inevitable. In Italy, the need for electronic health and artificial intelligence upsurged after SARS-CoV-2 dramatically hit the region and caused the death of numerous residents [28]. Bernardi et al. further insisted that the country-maintained follow-up medical cases, emergency procedures, and oncological surgeries [28]. In a similar perspective, Gironi et al. argued that the field of dermatologic surgery experienced numerous cancellations, thereby forcing patients to turn to eHealth for consultations and non-urgent processes [29]. Further, many hospitals in Italy have realized the need to adopt artificial intelligence as failure to integrate the notion has led to increased infections among staff and residents.

In Germany, the use of electronic health has increased during the SARS-CoV-2 pandemic, implying that future development is necessary and possible after the pandemic. For example, a study by Kirchberg et al. revealed that forty-two percent of the study participants, who were physicians, integrated medical applications in their phones [30]. However, the study showcased that 82 percent of the research respondents admitted that they lacked sufficient knowledge on a myriad of eHealth factors such as the legal issues and information safety of medical apps in addition to cloud computing properties. In the United Kingdom, the adoption of eHealth in surgery has increased due to the reduction of GP visits. A study carried out by Hutchings revealed that GP visits were at 80% before the pandemic, which dramatically shifted to 40% after the pandemic, as shown in Figure 3 [31]. These statistics reflect enhanced adoption of electronic health, and the trend is likely to continue after the pandemic as eHealth is efficient and effective. Moreover, the NHS introduced video consultations instead of face-to-face meetings with all patients to reduce the number of people visiting the hospital and the chances of transmitting the virus [21].

More importantly, France has demonstrated an increased use of electronic health during the pandemic, particularly in the field of colorectal cancer [32]. Priou et al. argued that the factor needed further evaluation and better execution to handle concerns revolving around delayed detection in surgery [32]. The use of artificial intelligence was less in the surgical domain in France, particularly lung transplant surgery. The lack of protective equipment and reduced lung donations has significantly contributed to reduced surgeries [33]. In summary, despite the application of eHealth and AI during the pandemic in surgery, the adoption rate was minimal to attain satisfactory results. However, there is room for improvement since Donell et al. declared that European countries are still developing guidelines to ensure the medical staff and patients are safe [34]. For instance, the European Hip Society and the European Knee Association have come together to develop eHealth and AI initiatives to deal with a future pandemic similar to SARS-CoV-2 [34]. Therefore, Europe is likely to enhance its adoption of electronic health and AI after the pandemic.

### 4.4. Development of Electronic Health and Artificial Health in Surgery after the SARS-CoV-2 Pandemic in Asia

According to Guo et al., in China, grown-ups with a higher socioeconomic status (SES) had higher eHealth literacy and received most of their information on SARS-CoV-2 through the Internet [35]. Correspondingly, artificial intelligence in surgery is evident, whereby Wuhan Wuchang Hospital utilized robotics by establishing an intelligent field comprising fourteen robots tasked with cleaning, delivering drugs, and measuring the temperatures of surgical patients [15]. In conjunction with that, Husain, Zhang, and Aung state that in health cases such as glaucoma in China, the health sector incorporated virtual consultation that reduced ocular morbidity from delayed care during the pandemic [36]. In addition, after the SARS-CoV-2 pandemic, health care sectors in China started using digital technologies such as artificial intelligence and smartphone-based apps to change the control of glaucoma. Additionally, Bokolo posited that China adopted eHealth platforms such as WeChat and hotline to deal with surgery patients [21]. Other hospitals in Wuhan incorporated innovative health tools, and big data analytics monitored remotely via observation cameras in Beijing [21]. Consequently, the National Telemedicine Centre of China developed an emergency eHealth consultation system geared toward managing and monitoring patients’ health. During the pandemic, the country has further applied the Kirkpatrick training model in the emergency surgery department [22].

Moreover, in South Korea, the administration uses information obtained from social media to collect useful eHealth metrics from profiling the history of patients in producing automated information for the residents [21]. It helps the country manage the conduct and cohabitation of citizens and presents extra measures that prevent the spread of the SARS-CoV-2 virus. Kim et al. stated that South Korea further minimized hospital infections by establishing a triage-based in-hospital management structure through a data link developed between the national immigration service and SNUBH’s BestCare [37]. The approach offers an automatic triage, testing visiting patients’ risk of SARS-CoV-2 through checking their underlined disease and their current immigration records to foreign countries. Sinha and Rathi reported that Korea uses artificial intelligence to predict if people infected with the virus survive [38]. The country achieves this by using machine learning with hyperparameters tuning, profound learning models, and an auto encoder-based method for valuing the effect of contrasting factors on the illness range and gauging the possibilities of survival for those in quarantine [38].

Sugawara, Murakami, and Narimatsu stated that in Japan, half of the population depend on the Internet to obtain medical information since the country does not have restrictions on websites that advertise clinical treatment [39]. It assisted in curbing the spread of SARS-CoV-2 by encouraging people to avoid travel when seeking medical attention and reducing congestion in hospitals. A review by Damodaran et al. revealed that the adoption of electronic health and artificial intelligence in India during the pandemic was low due to the negative implications of the pandemic on the country [40]. The minimal adoption is mainly associated with the region’s third-world status since it is still developing and facing many healthcare inequalities. In general, the surgery department in Asia ought to invest in studies concentrating on eHealth and AI. These studies might occur after the SARS-CoV-2 pandemic, as many Asian countries have learned about the significance of the concepts.

### 4.5. Development of Electronic Health and Artificial Health in Surgery after the SARS-CoV-2 Pandemic in the Middle East

In the Middle East, the onset of the SARS-CoV-2 pandemic has triggered the utilization of electronic health and artificial intelligence in surgery. In Saudi Arabia, eHealth adoption increased during the pandemic since 89 percent of the population use the Internet, and more than 95 percent of Saudi residents have access to smartphones [41]. The Ministry of Health of the state introduced the Tetamman app to allow healthcare professionals to deliver remote services to all patients, including those in the surgical domain. Further, Sharma and Ahmed claimed that Saudi Arabia had integrated many Internet of things applications to permit the diagnosis of the SARS-CoV-2 virus [42]. For instance, the country has significantly applied cloud computing through the utilization of sensor devices. Hence, the development of eHealth and AI in Saudi Arabia is likely to pick up after the pandemic.

A study carried out by Ting et al. showcased that many hospitals situated in the Middle East did not have the relevant resources needed to distinguish SARS-CoV-2 from common flu, thereby indicating an eHealth and AI insufficiency in the locality [43]. In a different view, Ferrara et al. stated that the virus’s rapid spread prompted countries in the Middle East to adopt strict containment measures to avoid spreading the virus [44]. For example, in Lebanon, there have been numerous measures to promote eHealth adoption among refugees that are from Syria [45]. Talhouk et al. revealed that challenges such as failure to provide medical services to refugees have significantly hindered eHealth incorporation among Syrian refugees in Lebanon [45]. Subsequently, eHealth and artificial intelligence adoption in Syria cannot occur, given the increased levels of conflict in the locality. Bowsher et al. insisted that although the deployment of electronic health has increased during the pandemic, researchers still need to clarify the global norms applicable in the worldwide realm to ensure the prevalence of eHealth and AI in tumultuous regions [46].

Most importantly, Tara et al. argued that many electronic health advancements occurred during the pandemic, particularly in surgical procedures in Palestine [47]. For instance, through the ministry of health, the country established a system geared toward monitoring the health of residents and identifying the various hotspots for the spread of the SARS-CoV-2 virus [47]. Another review by Burney et al. revealed that Oman integrated eHealth during the pandemic by introducing the Tarassud Plus telephone application [48]. Consequently, Hassan, Rabbani, and Abdulla argued that Kuwait was the best-equipped country as it has many hospital beds and medical personnel [49]. On the contrary, Yemen has encountered the negative impacts of the pandemic due to the fragility of the system evident through the absence of protective equipment and minimal medical supplies in the surgical field [49]. Despite the increased adoption of electronic health applications in the Middle East, the success of the apps and subsequent AI is far from attaining. The statistics above imply that the whole Middle-Eastern region will adopt relevant programs to develop eHealth and artificial intelligence after the pandemic. However, embracing the concepts may take longer for countries such as Libya and Syria since they face political conflicts.

### 4.6. Development of Electronic Health and Artificial Intelligence in Surgery after the SARS-CoV-2 Pandemic in Africa

A study conducted by Ogundele et al. revealed the necessity of developing electronic health in pediatric surgery after the SARS-CoV-2 was necessary. There was a termination of elective surgeries in Nigeria and an enormous decline in emergency surgeries performed on toddlers [50]. The reason why surgeons are not attending to their duties is that they do not feel safe operating on patients, especially those who have contracted SARS-CoV-2. Furthermore, people in Nigeria are currently living in fear of contracting the various, as it has affected their lives negatively and has also made the death toll rise. Further, Kelechi et al. argue that if technology and health care are linked, it will de-escalate the rate of SARS-CoV-2 infections and save more people [51]. Additionally, the technology spreads awareness of the disease and assists the Government in monitoring and giving accurate compliance with the lockdown measures. According to Elkhouly, Salem, and Haggag, telemedicine in Egypt has helped reduce the rate of infection for a specific cluster of cancer patients [52]. Treatment schedules were rescheduled to give patients seeking surgeries related to cancer priority to treatment.

Saba and Elsheikh stated that in Egypt, artificial intelligence models such as NARANN and ARIMA are used to assist in forecasting the prevalence of COVID-19, whereby the predicted information has a high determination coefficient for all groups of data [53]. Further, Tara et al. reported that eHealth adoption in Egypt prompted the triaging of patients into three distinct categories [47]. The telemedicine care division entailed using electronic channels to ensure patients received doctors’ necessary care [47]. In Algeria, eHealth and artificial intelligence adoption are not successful due to the dysfunctional healthcare system adopted in the region [47]. A review by Owoyemi et al. revealed that the eHealth and AI adoption in the African continent encounters the challenge of unavailability of large clinical datasets to steer the training of artificial intelligence models [54]. In addition, legal issues concerning the use and privacy of eHealth and AI tools have significantly made it challenging for Africa to integrate the concepts into its healthcare domain so far.

In Morocco, the adaptation of electronic health in surgery is evident via the adoption of the National Physicians Order telehealth program to reduce unnecessary hospital visits during the pandemic [47]. In addition, the country has developed a legal framework for ensuring the safe delivery of telemedicine services, thereby encouraging the further development of the concept in surgery after the pandemic [47]. Moreover, in Ethiopia, AI in surgery is evident as the county launched high-profile artificial intelligence programs in the region. For instance, the thirty and one hundred and thirty universities and polytechnics respectively conduct technological studies [52]. Despite the prominence of eHealth and AI in Africa, numerous analyses and efforts ought to ensure that counties have the proper mechanisms to deal with the challenges to AI adoption in surgery. These studies are likely to increase post- pandemic since many countries have learned their lessons through the shortcomings of SARS-CoV-2.

## 5. Summary of Evidence

The application of artificial intelligence and eHealth in surgery remains a critical issue that needs more studies and developments. A survey by Hashimoto et al. revealed that the global adaptation of AI in surgery largely depends on technological developments such as mobile phones and cloud computing [55]. Attempts to develop robotic surgery are evident through a project by John Hopkins University, where the students created the smart tissue autonomous robot (STAR) that contained algorithms geared toward replacing outperforming human surgeons in procedures involving animals. Hashimoto, Ward, and Meireles further insist that AI utilization in surgical procedures is narrow and remains essential for all surgeons to approach the issue with a healthy skepticism to ensure successful integration [56].

Currently, the development of electronic health and artificial intelligence encounters numerous implementation shortcomings. For example, in telesurgery, challenges such as latency time and delivery of feedback are prevalent. The solution to such problems requires the integration of enhanced technologies such as the Internet of things and 5G networks. Robotic surgery reflects a minimally invasive surgical procedure that uses computer-aided robot technologies to allow surgeons to complete surgical processes with minimal complications. The primary challenge of using robots in surgery arises from safety issues revolving around confidentiality assurance. Since robotic surgery occurs online via computer networks, the risk of a data breach is significant, and the protection of patient data is vital to prevent any legal claims against hospitals and surgeons. Further, the incorporation of AI in surgery faces drawbacks primarily associated with patient safety. For instance, the association between robotic procedures and patient death has prevailed, despite arguments that physicians’ inexperience with the robots might have contributed to the occurrences. In addition, the high cost of robots has hindered their adoption in surgery, especially in low-income states. The pandemic has further necessitated the need for enhanced information security as the engagement of people online spiraled and cyber security risks increased.

## 6. Conclusions

Electronic health is a recent concept that reflects the use of technology to deliver healthcare services. Many studies in the field are yet to reach a consensus on the primary definition of the concept. Artificial intelligence is older than eHealth, and it represents the use of machines to enhance the delivery of care in the medical realm. Presently, the application of eHealth and AI has increased, mainly due to the onset of the SARS-CoV-2 pandemic. The primary challenge in the research area is the unavailability of substantial research focusing on eHealth and AI development in surgery in developing countries such as some states in the Middle East, Asia, and Africa. Another challenge in the subject area is the absence of adequate data on the various shortcomings of adopting AI in surgery. Currently, the solution of the two issues remains pending despite many researchers encouraging fellow scientists and medical practitioners to increase studies on AI challenges in surgery, particularly in developing nations. The lessons learned from previous attempts on the topic include that surgeons should collaborate with researchers to develop efficient and inexpensive robots.

## Figures and Tables

**Figure 1 jcm-10-04789-f001:**
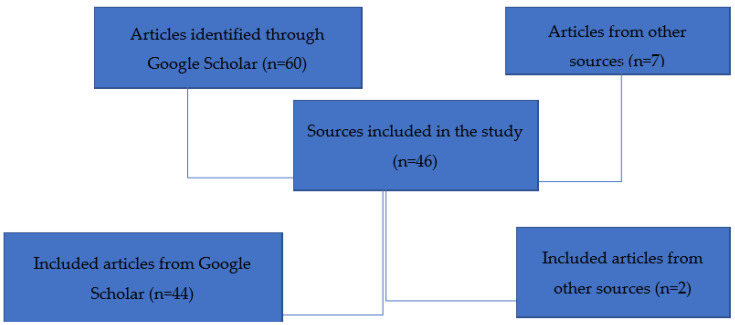
Selection of sources for the study.

**Figure 2 jcm-10-04789-f002:**
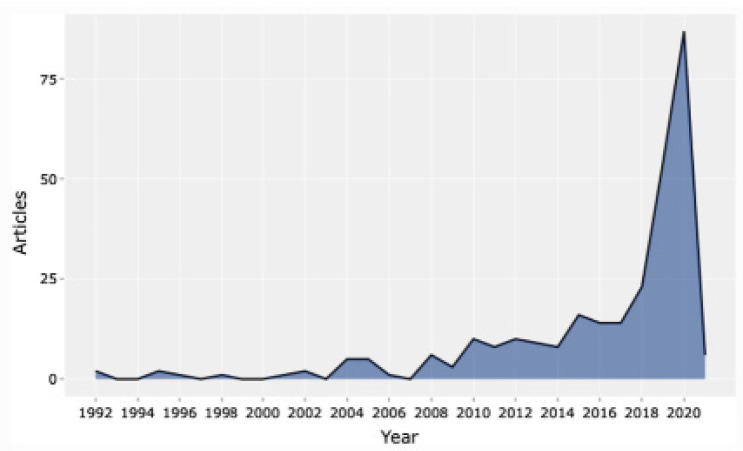
Articles published on artificial intelligence from 1992–2020 (permission to reproduce from Secinaro et al. [27]).

**Figure 3 jcm-10-04789-f003:**
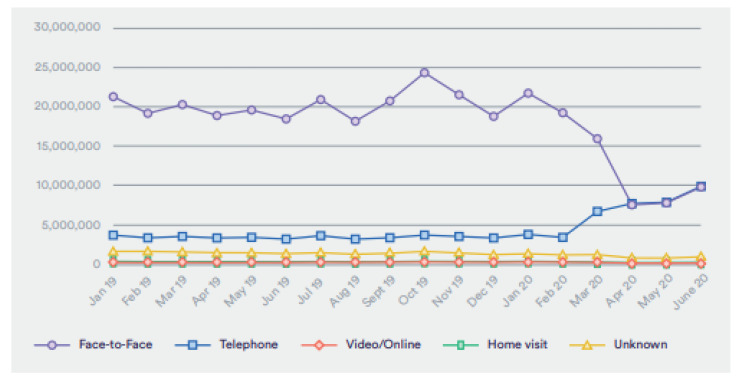
Appointments to the general practitioners from January 2019 to June 2020 (permission to reproduce from Hutchings [31]).

## Data Availability

The datasets used and/or analyzed during the current study are available from the corresponding author on reasonable request.

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
