# Peer review of "The Development of Electronic Health and Artificial Intelligence in Surgery after the SARS-CoV-2 Pandemic—A Scoping Review"

_jcm, 2021, doi:10.3390/jcm10204789_

Round 1

Reviewer 1 Report

This paper presents an interesting idea. However, it is a bit complicated for the reader to follow. To improve this, I think it would be necessary to improve the introduction. It should indicate what the intention of the paper is, how the papers reviewed have been selected and what literature review methodology has been followed to ensure the quality of the review. In the review it is not clear in which of proposals Artificial Intelligence (AI) is used and for what purpose. In other words, the concept of AI is a very broad concept and in eHealth it can be used for many different purposes. For instance, in line 237 it says "For example, in Israel, the government passed a law that allowed it to track and monitor the telephone data of all the patients suspected to have the virus", how does this relate to AI? This is simply tracking the movements of specific patients, which, per se, is not related to AI. This problem can be found in other proposals mentioned in the paper. Finally, the conclusions are a little prosaic and do not provide new information in the area. In a literature review, a more in-depth analysis should be obtained that provides the reader with an understanding that goes beyond a mere summary of the articles reviewed. Some of the questions that could or should be answered are: what are the main challenges in the subject area? How have they been solved? Which ones remain to be solved? What are the success/failure cases or lessons learned? Minor comments I think that the figures should be a little smaller in size and especially, Figure 2 should be graphically improved, it presents poor quality.

Author Response

我也改进了介绍和结论。此外,我在文献综述中添加了一些段落来回答您指出的问题。此外,我提高了图 2 的质量。关于第 237 行和其他建议,我没有明确谈论人工智能,因为我在该部分的主要重点是电子健康。我将讨论分为外科电子健康和外科人工智能。

Reviewer 2 Report

Dear Authors, 
Congratulations on your prepared manuscript.
The thematic scope of the research findings presented is coherent and consistently leads to the conclusion.
The subject matter is generally taken up in the literature, however both the tools and methodology presented in the article I assess as original. A certain discomfort remains the lack of more advanced analytical tools used.

Author Response

I advanced the analytical tools used through the use of the PRISMA approach.

Reviewer 3 Report

This manuscript is a review on the development of eHealth and AI in surgery, and putting emphasis on how SARS-CoV-2 pandemics promoting the needs and adoption of these usage. The authors reviewed the published literature, and presented the trend and application area by area in the world, and analyzed the potential opportunities and threats in and after the pandemics. Generally, this review provides comprehensive information and useful message for medical professionals. I have one concern on the manuscript hoping the authors can address it further.

Concern:

1. Since the main theme of this review is on surgery, so I expect the authors can discuss the issue in more details, such as tele-surgery, the robot application and the safety issues regarding patient safety, confidentiality reassurance, and development of information security during the pandemics and the era after the pandemics.

Author Response

我通过在讨论结束时强调人工智能在手术中的缺点来解决这个问题

Reviewer 4 Report

Manuscript ID jcm-1420406, titled: "The Development of Electronic Health and Artificial Intelligence in Surgery after the SARS-CoV-2 Pandemic-a Literature Review," focuses on the empowerment of the concepts of electronic health and artificial intelligence, which mainly depend on the efforts of countries to advance the notions of surgery.

To improve your work, I suggest:

  1. improve the title, indicate that it is a scoping review.
  2. This review does not have so much depth, so it should be treated as a scoping review.
  3. I suggest applying a method for scoping review, review the PRISMA method for scoping review http://www.prisma-statement.org/Extensions/ScopingReviews
  4. In the Abstract, include a paragraph of the contribution of this research, limitations and future work.
  5. Check the English grammar and spelling of the entire paper.
  6. Indicate whether this research uses publicly available information or has a data set with access to the data for researchers to replicate the experiment.
  7. To make your research replicable, I suggest placing the study data in a dataset; you can create one at https://data.mendeley.com/.
  8. Improve the quality and resolution of Figure 1.
  9. Improve the quality and resolution of Figure 2.
  10. Shorten the introductory section; it is too long and repetitive.
  11. Explain the method better; if possible, include a sketch of the applied method.
  12. Improve the discussion section.
  13. Improve the conclusions section.
  14. The authors should work a little harder to improve their review.
  15. I encourage them to review other similar works to improve their article.

Author Response

我改进了标题并应用了 PRISMA 方法。此外,我校对了论文以消除任何错误并增强了摘要。此外,我提高了图像的分辨率并缩短了介绍。我还提供了所采用方法的草图,由于该研究使用了公开可用的数据,因此我没有将其放入数据集中。尽管如此,我改进了讨论和结论部分。

Round 2

Reviewer 1 Report

The changes made seem to me to be appropriate.

Reviewer 4 Report

I congratulate the authors for their work. With the improvements applied, the work is ready for publication.